# Normalizing Effect of Heat Treatment Processing on 17-4 PH Stainless Steel Manufactured by Powder Bed Fusion

**Si-Mo Yeon** [1,2], **Jongcheon Yoon** [1], **Tae Bum Kim** [1], **Seung Ho Lee** [3], **Tea-Sung Jun** [2], **Yong Son** [1] and **Kyunsuk Choi** [4,*]

1. Advanced Joining & Additive Manufacturing R&D Department, Korea Institute of Industrial Technology, Siheung 15014, Korea; simo@kitech.re.kr (S.-M.Y.); jongcheon89@kitech.re.kr (J.Y.); xoqja78@kitech.re.kr (T.B.K.); sonyong@kitech.re.kr (Y.S.)
2. Department of Mechanical Engineering, Incheon National University, Yeonsu, Incheon 22012, Korea; t.jun@inu.ac.kr
3. HYU-KITECH Joint Department, Hanyang University, Ansan 15588, Korea; qhtmho@hanyang.ac.kr
4. Department of Industry-University Convergence, Hanbat National University, Yuseong, Daejeon 34158, Korea
* Correspondence: kschoi@hanbat.ac.kr; Tel.: +82-42-828-8586; Fax: +82-42-821-1585

**Abstract:** Laser powder bed fusion (L-PBF)-processed 17-4 PH stainless steel (SS) generally exhibits a non-equilibrium microstructure consisting mostly of columnar δ-ferrite grains and a substantial fraction of retained austenite and martensite, contrary to 17-4 PH SS wrought with a fully martensite structure and coarse grains. Despite the different microstructures of L-PBF and wrought 17-4 PH SS, post-processing is typically performed using the conventional heat treatment method. The insufficient effect of the heat treatment on the L-PBF product produces a δ-ferrite phase in the microstructure. To obtain improved mechanical properties, the addition of a normalizing treatment to the conventional heat treatment after L-PBF in a nitrogen gas environment was investigated. The fully martensitic matrix developed by adding the normalizing treatment contained homogeneous Cu precipitates and exhibited a similar or improved strength and elongation to failure compared to the wrought SS.

**Keywords:** 17-4 PH stainless steel; selective laser melting; laser powder bed fusion; heat treatment; microstructure; mechanical properties

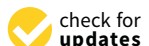



## 1. Introduction

A 17-4 PH stainless steel (SS) is a chromium-nickel-copper precipitation-hardenable alloy that is widely used in the aerospace, marine, power generation, chemical plant, and medical industries. It requires high strength and corrosion resistance against thermal exposure to temperatures of approximately 316 °C [1,2]. When fabricated using the conventional process, this alloy is predominantly composed of a martensite phase and is strengthened by heat treatment (aging treatment at 482 °C for 1 h after solution treatment at 1060 °C) to precipitate Cu particles in the martensite [3,4]. Recently, higher value-added industries, such as aerospace, health care, power generation, and automobiles, have demanded more design freedom and complex manufacturing technology to improve functions and performance [5,6].

Additive manufacturing technology has attracted attention as a global solution, with the functional enhancement of dissimilar materials, such as superior corrosion resistance, excellent thermal stability, high hardness, better wear performance, [7–9], and the fabrication of near-net-shape parts with higher dimensional accuracy [10].

A 17-4 PH SS is comparatively difficult to work with and has poor machinability, thereby leading to a demand for additive manufacturing technology for the fabrication of 17-4 PH SS products, with customized shapes for specialized applications [11]. Previous studies have shown that the characteristics of laser powder bed fusion (L-PBF) products are strongly dependent on the materials (powder composition, initial atomizing media, etc.),

the build-chamber atmosphere, the design parameters, the process conditions, and the post-heat treatments [12].

Additively manufactured 17-4 PH SS has a unique microstructure that affects its mechanical properties and phase evaluations during post-processing. It exhibits an anisotropic microstructure with columnar grains that are oriented parallel to the build direction and fine grains with average sizes smaller than 10 μm when cooled quickly at $10^5$–$10^7$ K/s [13]. It also contains non-equilibrium microstructures comprising mixed phases of δ-ferrite (BCC), austenite (FCC, γ), martensite (BCT, α′), and carbide precipitations [11].

The presence of retained austenite and δ-ferrite phases and copper precipitates in the martensite matrix and the changes in grain size that occur during the manufacturing process significantly influence the mechanical properties of the fabricated parts. The austenite phase reduces the amount of copper precipitated during the aging step because of the higher solubility and lower diffusivity of copper in austenite than in martensite [14–16]. The remaining ferrite also exhibits delayed hardening behavior [17,18]. The fraction of austenite and ferrite remaining after the L-PBF process and heat treatment depends on the build-chamber environment ($N_2$ or Ar gas) [19], grain size (which may be reduced by using a high cooling rate) [20,21], residual stress in the grains or at the grain boundaries [19,22,23], composition (chromium-to-nickel equivalent value) [24], and the presence of austenite stabilizers such as C and N [25].

Figure 1 shows the mechanical properties of the as-built (AB) specimens produced by the L-PBF and heat-treatment (HT) specimens collected from the literature [23,26–30] and the detailed data of mechanical properties in Figure 1 is presented in the Supplementary Table S1. The graph shows that the yield strength of the AB L-PBF parts is less than 800 MPa, and the yield strength and elongation of the HT parts are less than the corresponding values of the wrought parts, for which ASTM A693 (H900 condition) is the standard reference [31]. The reduced yield strength of the L-PBF 17-4 PH SS products is important because it is used to determine the safety coefficient in most structural designs. The mechanical properties of the L-PBF products should be improved, and further research on additive processing and post-process optimization is required.

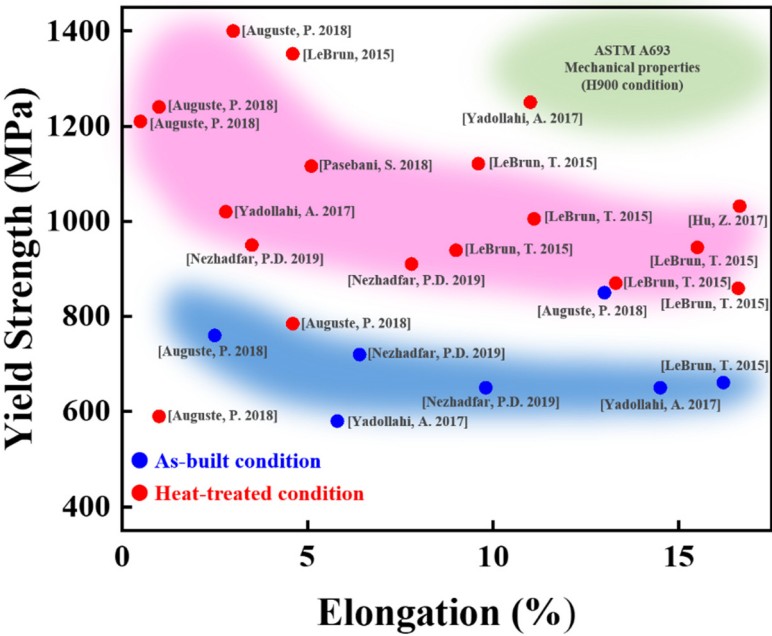

**Figure 1.** Mechanical properties of 17-4 PH SS produced by L-PBF from previous studies [23,26–31].

Further research is required to extend the industrial application of the L-PBF process because it is not sufficiently economical in many cases. The manufacturing cost of the additive manufacturing process exceeds that of most traditional methods [32–36]. In

previous studies, the effects of the mechanical properties of Ar and N atmospheres during the L-PBF process were studied, and it is generally reported that poor mechanical properties appear in a nitrogen atmosphere [12,15,19].

Motivated by this, we studied the microstructure and mechanical properties based on the additional heat treatment and heat treatment steps of 17-4 PH SS fabricated under nitrogen conditions, which is advantageous in terms of cost, although it has unfavorable mechanical properties. Therefore, a heat treatment method for obtaining a stable microstructure and predictable properties is proposed.

## 2. Materials and Methods

### 2.1. 17-4 PH SS Powder

Gas-atomized 17-4 PH SS powders (obtained from 3D-systems Inc., Rock Hill, SC, USA) with particle sizes ranging between 5–30 μm and with an average diameter of 10.63 μm were used in this study, as shown in Figure 2. The chemical composition of the powders is listed in Table 1. Inductively coupled plasma-atomic emission spectroscopy (ICP-AES) and carbon-sulfur analysis confirmed that the 17-4 PH SS powder had a nominal chemical composition that conformed to the 17-4 PH SS specification (ASTM A693-16).

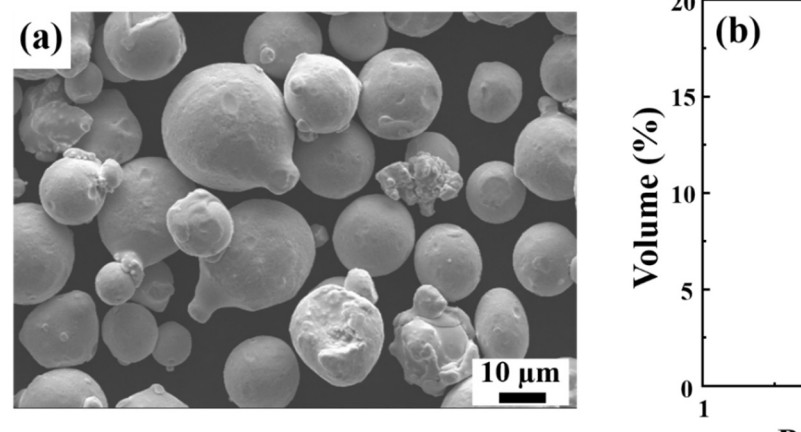 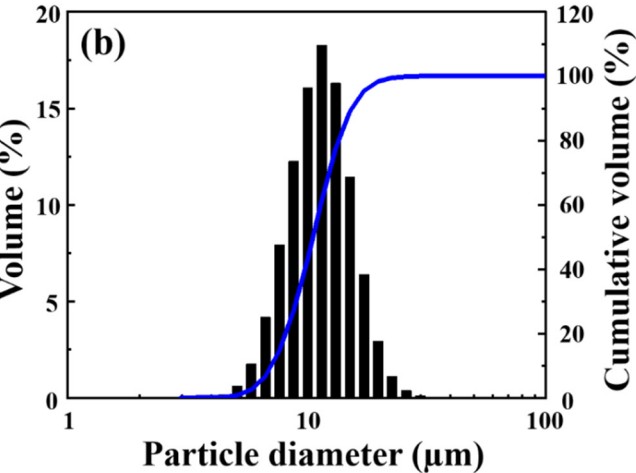

**Figure 2.** (**a**) SEM image and (**b**) particle size distribution of 17-4 PH stainless steel powder.

**Table 1.** Chemical compositions (wt.%) of 17-4 PH stainless steel powder.

| Fe | Cr | Ni | Cu | Si | Mn | Nb | P | S | N | C |
|-----|-------|------|------|------|------|------|-------|-------|-------|-------|
| Bal. | 17.10 | 4.40 | 4.42 | 0.92 | 0.85 | 0.24 | 0.021 | 0.009 | 0.009 | 0.027 |

### 2.2. Part Fabrication by L-PBF

The 17-4 PH SS test specimens were produced using a 3D-systems ProX300 machine. The specimens were fabricated using a custom-built method, as shown in Figure 3a. The laser scanning strategy for all the specimens consisted of an outer contour area and an inner hatching area. The contour was scanned in two lines at intervals of 40 μm, and the hatching in a 10 mm regular hexagonal pattern overlapped the remaining area by 50 μm, as shown in Figure 3b. All specimens were fabricated with the same parameters: laser power of 135 W, laser scan speed of 1200 mm/s, layer thickness of 40 μm, and hatching distance of 50 μm. The oxygen content in the build chamber was reduced to 1000 ppm to prevent specimen contamination. Nitrogen gas was used as the inert instead of argon gas, for environmental control of the build chamber during the L-PBF process.

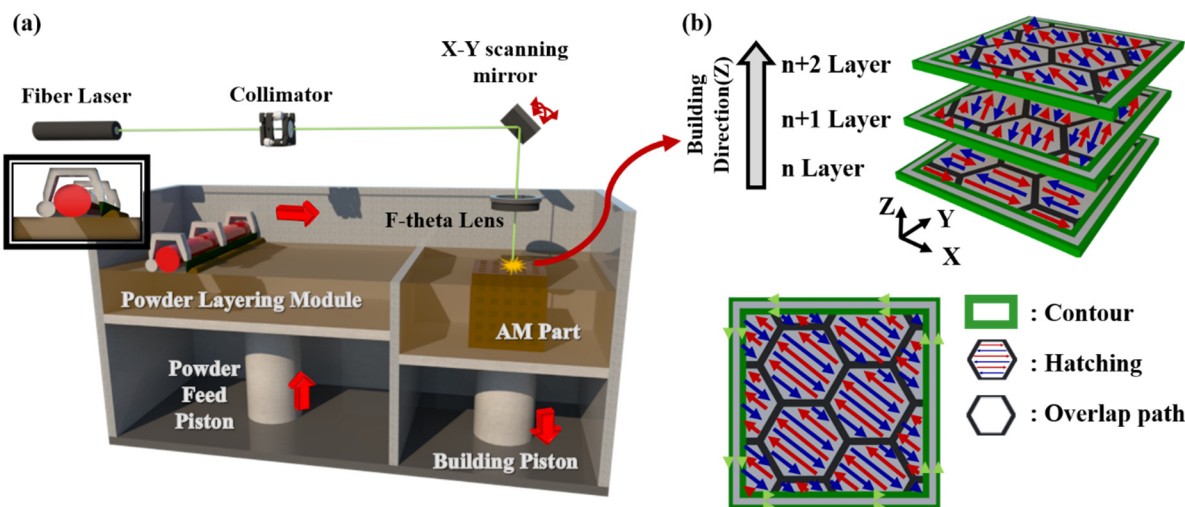

**Figure 3.** Schematic of (**a**) L-PBF machine and (**b**) strategies of laser scanning and layering.

### 2.3. Post-Heat Treatment Procedures

The details of the post-heat-treatment procedures used in this study are listed in Table 2. To analyze the effects of thermal exposure on microstructure and phase evolution, the specimens were heat treated using conventional methods (ASTM A693, Solution A + H900, aging treatment at 482 °C for 1 h, and after solution treatment at 1060 °C) and subjected to normalizing treatment (1200 °C for 4 h before the solid solution treatment). Five types of heat-treated specimens subjected to different numbers of treatment steps were prepared.

**Table 2.** Heat treatment condition of sample using fabricated L-PBF.

| Sample Name | Heat Treatment Condition |
|---|---|
| As-built (AB) | - |
| Normalizing treatment (N) | 1200 °C, 4 h + Furnace cooling |
| Solution treatment (S) | 1060 °C, 1 h + Gas cooling |
| Solution + Aging treatment (SA) | 1060 °C, 1 h + Gas cooling → 482 °C, 4 h + Air cooling |
| Normalizing + Solution + Aging treatment (NSA) | 1200 °C, 4 h + Furnace cooling → 1060 °C, 1 h + Gas cooling → 482 °C, 4 h + Air cooling |

### 2.4. Characterizations

Tensile tests were performed on the specimens produced based on the specifications of round bar specimen no. 3 in the ASTM E8 standard, using a universal testing machine (Landmark, MTS, Eden Prairie, MN, USA), with hydraulic wedge grips and an extensometer. Each specimen had a gauge length of 25 mm and a diameter of 6 mm. Tensile tests were performed at room temperature at a specific crosshead speed of 2 mm/min. The obtained data were analyzed to plot the stress–strain curves and determine the 0.2% proof yield stress, ultimate tensile strength, and elongation to failure. X-ray diffraction (XRD) analysis was performed using a Cu K$\alpha$ source (40 kV and 30 mA) to evaluate the phase composition over a 2θ range of 20–90°. Microstructural observations were conducted using optical microscopy (OM, Vert A1, Carl Zeiss, Oberkochen, Baden-Württemberg, Germany) and scanning electron microscopy (SEM, JSM-F100, JEOL, Akishimam, Tokyo, Japan). The specimens for electron backscattered diffraction (EBSD) analysis were polished using ion milling (cross-section polisher IB-19520CCP, JEOL, Akishimam, Tokyo, Japan) at 3 kV for 2 h. EBSD analysis was performed using an SEM and EBSD system (Velocity super, EDAX, Pleasanton, CA, USA) with an acceleration voltage of 20 kV. To determine copper precipitation, a transmission electron microscope (TEM, Talos F200X model, FEI, Hillsboro, OR, USA) was used at 200 kV.

## 3. Results and Discussion

### 3.1. Mechanical Properties

The effects of the heat treatments on the mechanical properties of L-PBF 17-4 PH SS were evaluated using tensile tests. To obtain reliable and reproducible data, five tests were performed for each HT condition. The results are shown in Figure 4.

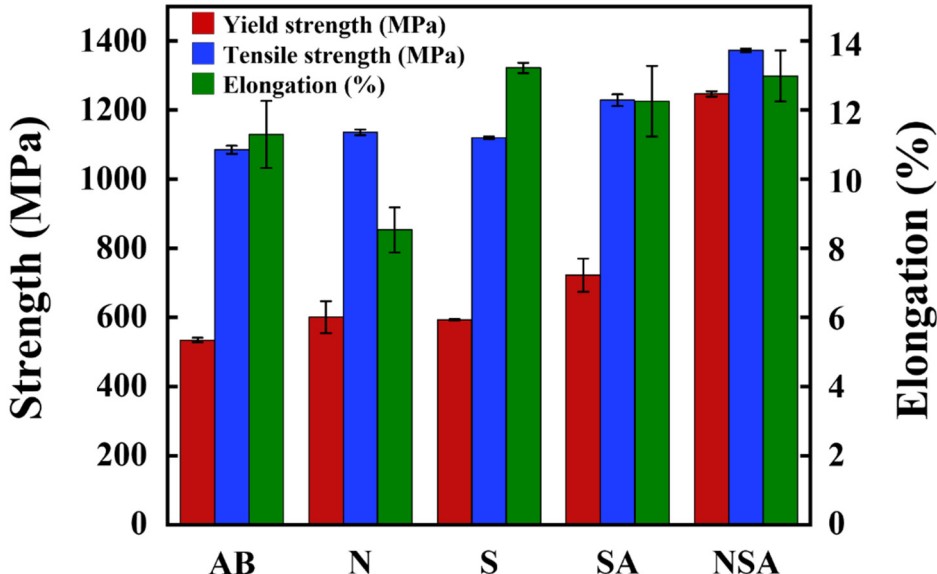

**Figure 4.** Yield strength, tensile strength, and elongation for the L-PBF as-built and heat-treated 17-4 PH sample.

The tensile strengths of the AB and SA specimens treated using the conventional heating method were similar to those reported by other researchers, as shown in Figure 1 [23,26,27]. Conversely, the tensile strength of the NSA specimen significantly exceeded that of the 17-4 PH SS manufactured by the L-PBF process by other researchers and met the requirements of the H900 condition in the ASTM A693 standard. These results indicate that the mechanical properties were significantly dependent on the application of normalizing treatment prior to the solution and aging treatments.

The yield strength of the AB specimen obtained in this study was 535 MPa, which is the lowest among those of the specimens obtained under all the HT conditions and lower than that of the L-PBF 17-4 PH SS produced by other groups. The yield strengths of the S and SA specimens were 593 MPa and 722 MPa, respectively, which were slightly higher than those of the AB specimen but significantly lower than those of the H900 HT-wrought 17-4 PH SS or the H900 condition of the ASTM A693 standard, as shown in Figure 1.

Conversely, the yield strength and elongation of the NSA specimen significantly increased to 1265 MPa and 12.95%, respectively, and the standard deviation decreased compared to that of the SA specimen, indicating the uniformity of the mechanical properties in the NSA specimen. These results indicate that microstructural changes occurred during the normalizing treatment, affecting the microstructural evolution during the solution and aging treatments.

### 3.2. XRD Patterns and Optical Micrography Images

XRD analysis was performed to determine the phases and microstructures of the AB and HT specimens. As shown in Figure 5, the AB and N specimens were composed of a large fraction of body-centered cubic (BCC) and certain amount of face-centered cubic (FCC) phases, whereas the S, SA, and NSA specimens consisted almost entirely of the BCC phase. This implies that during cooling after the solution HT, the FCC phase in specimens AB and N was transformed into the BCC phase.

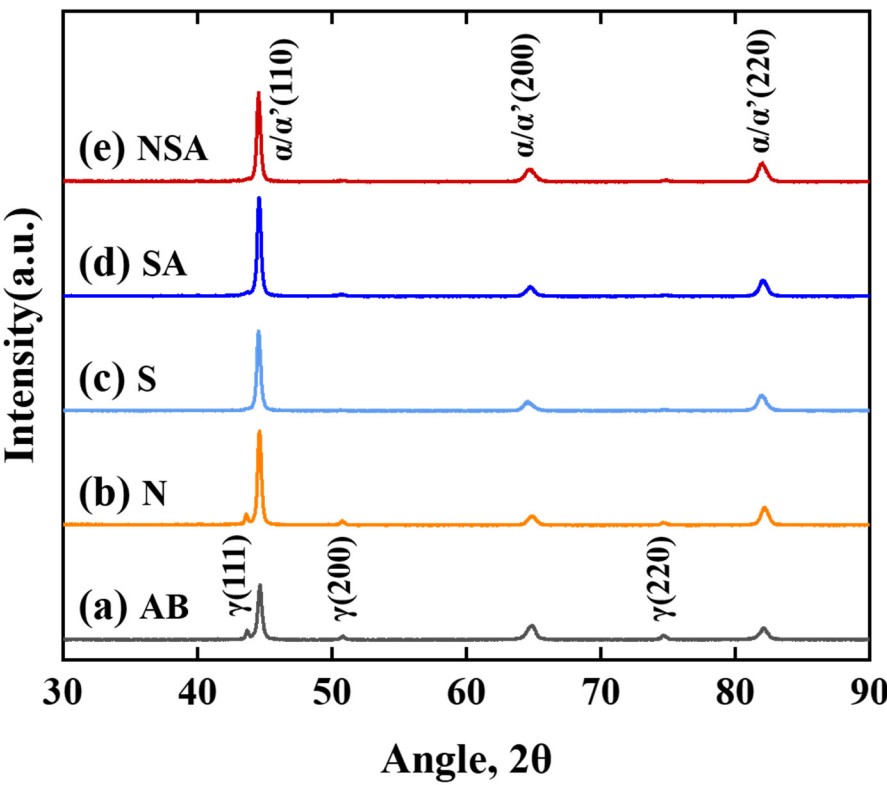

**Figure 5.** XRD patterns of the as-built and heat-treated 17-4 PH samples.

The SA and NSA specimens showed no significant differences in their XRD results despite exhibiting large differences in their tensile and yield strengths, as shown in Figure 4. This could be because the body-centered tetragonal (BCT) martensite and the BCC δ-ferrite were significantly similar, so there were not distinguishable in the XRD pattern. This is because of the fact that the BCT martensite crystal structure in 17-4 PH SS is the BCC phase with a small amount of lattice distortion, similar to the distortions seen in significantly low-carbon SS [14].

To observe the changes in the microstructures, optical micrography images of the cross-sections parallel to the build direction of the AB, N, S, SA, and NSA specimens were examined. The AB specimen exhibited the typical microstructure of L-PBF 17-4 PH SS, with large, elongated grains grown across melt pools along the build direction and equiaxed fine grains at the melt pool boundaries. The large elongated and equiaxed grains in Figure 6a are reported as δ-ferrite and retained austenite, respectively [37].

The phase fraction of the AB L-PBF 17-4 PH SS depended on the $Cr_{eq}/Ni_{eq}$ value of the initial powder [24,25]. The $Cr_{eq}$ and $Ni_{eq}$ values were calculated using WRC-1992 equation [38]:

$$Cr_{eq}(wt.\%) = Cr + Mo + (0.7 \times Nb) \tag{1}$$

$$Ni_{eq}(wt.\%) = Ni + (35 \times C) + (20 \times N) + (0.25 \times Cu) \tag{2}$$

High levels of δ-ferrite (80–95%) were obtained when the $Cr_{eq}/Ni_{eq}$ ratios were between 2.65 and 2.81, and nearly 25 vol.% of δ-ferrite appeared when the $Cr_{eq}/Ni_{eq}$ ratio was less than 2.36 [24]. The $Cr_{eq}/Ni_{eq}$ value of the powder used in this study was 2.60 (using WRC-1992 equations), and a relatively high δ-ferrite fraction was induced.

After the heat treatment, the columnar δ-ferrite, fine martensite, and austenite phases in the AB specimen were reconstructed into equiaxed austenite grains through recovery and recrystallization. When only solution heat treatment was performed, the recrystallized austenite grains were smaller on average and irregular in size, ranging from several micrometers to several tens of micrometers (Figure 6c,d). When the specimens were subjected to normalizing HT maintained at 1200 °C for 4 h, austenite grains with sizes

30–50 μm were uniformly present owing to grain growth (Figure 6b,e). The Hall–Petch relationship implies that the strength and elongation to failure of a material are enhanced at smaller grain sizes. The optical micrography results suggest that the SA specimen has greater strength and elongation than the NSA specimen. However, an opposite result was obtained, as shown in Figure 4.

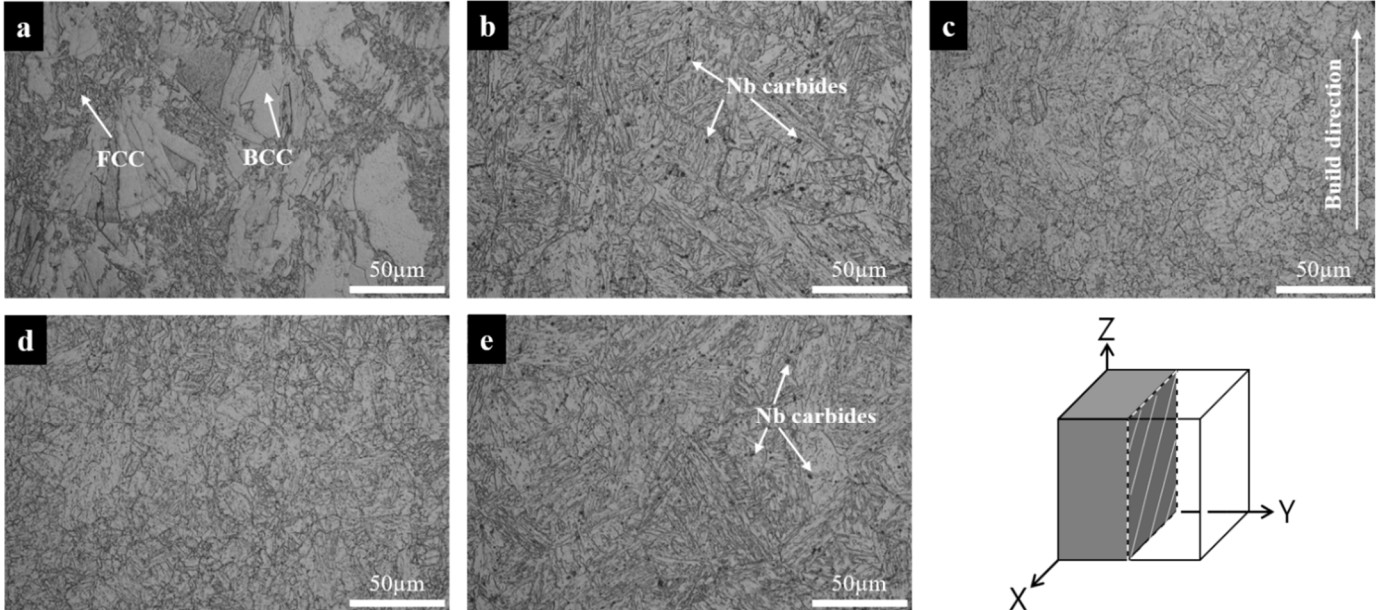

**Figure 6.** Optical micrographs obtained from etched sections of the as-built and heat-treated 17-4 PH samples: (**a**) AB, (**b**) N, (**c**) S, (**d**) SA, (**e**) NSA.

The normalized specimen included several micrometer-sized spherical particles, which were verified to be Nb-rich carbides by EDS (Figure 6b). The Nb-rich carbides in the N specimen became smaller in the NSA specimen matrix because of decomposition after the post-solution and aging treatments (Figure 6e).

### 3.3. Phase Analysis

Further EBSD analysis was performed to investigate the phases of the AB and HT specimens in detail. Figure 7 shows a cross-sectional view of the AB and HT specimens along the build direction. The orientation and grain boundaries are revealed more clearly than those in Figure 6.

The AB specimen microstructure consisted of columnar grains (tens of micrometers in size) and fine-equiaxed grains preferentially located between the columnar grains, as shown in Figure 7a. The columnar grains were a fully δ-ferritic BCC microstructure formed by an extremely high cooling rate of $10^5$–$10^7$ K/s, and the fine-equiaxed grains were a mixture of BCC martensitic laths and fine FCC austenite grains. These results are similar to those obtained in a previous study on an AB 17-4 PH SS fabricated using L-PBF [11].

Figure 7b,d shows the microstructures of the specimens after HT. The grain-size differences with and without normalization in the treated specimens are consistent with the results from the optical images. The microstructures of the normalized specimens shown in Figure 7b,e are relatively coarse and homogeneous, whereas those without normalization (Figure 7c,d) are smaller and contain some coarse grains.

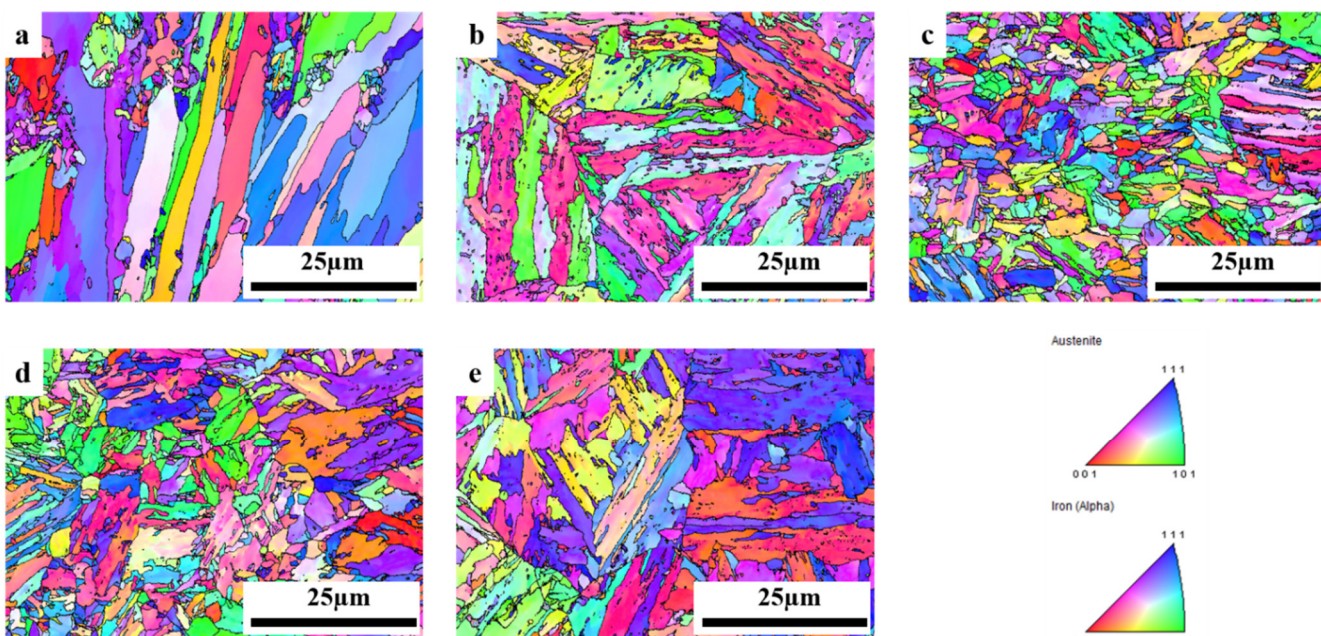

**Figure 7.** EBSD orientation maps obtained from the as-built and heat-treated 17-4 PH samples: (**a**) AB, (**b**) N, (**c**) S, (**d**) SA, (**e**) NSA.

The phase fractions of all the post-HT specimens were similar, consisting mostly of the ferrite phase and approximately 1% austenite, except for the N specimen, which contained approximately 10% austenite (each phase map and fraction of all specimens is shown in Supplementary Figure S1 and Table S2). This result is contradictory to that of a previous study, in which the incomplete transformation of martensite occurred more easily when nitrogen gas was used instead of argon gas as the build-chamber environment [12]. In this study, gas cooling was used instead of air cooling in the solution treatment process to avoid the retained austenite phase and obtain a full martensite phase. As expected, a significantly small fraction of the austenite phase was found in the specimens treated by gas cooling, as shown in Figure 7c,e.

The EBSD results shown in Figure 7d,e cannot explain the difference between the mechanical properties of the SA and NSA specimens, as shown in Figure 4. Additional analyses were performed to identify the cause and mechanism for this result.

### 3.4. Cu Precipitation

Age hardening is an HT process for increasing the yield strength through the uniform precipitation of 10–20 nm Cu particles and inhibiting the dislocation movement through the Orowan mechanism. The yield strengths of the 17-4 PH SS specimens were significantly affected by the amount of Cu precipitation. Therefore, TEM and EDS mapping analyses were performed to verify the correlation between the microstructure and the mechanical properties of the SA and NSA specimens after aging, as shown in Figure 8.

Nanoscale Cu-rich precipitates were observed in both the SA (Figure 8a) and the NSA (Figure 8d) specimens (results of the TEM-EDS map about the distribution of major compositions are shown in Figures S2–S6). The SA specimen displayed two distinct types of regions: Figure 8b shows no precipitation or only precipitates inhomogeneously, and Figure 8c shows Cu particles precipitated homogeneously. Figure 8a was magnified by 2.5 times in Figure 8b,c to clearly show the differences between the two regions. Homogeneous Cu precipitation was obtained after the normalizing treatment, as shown in Figure 8d,e.

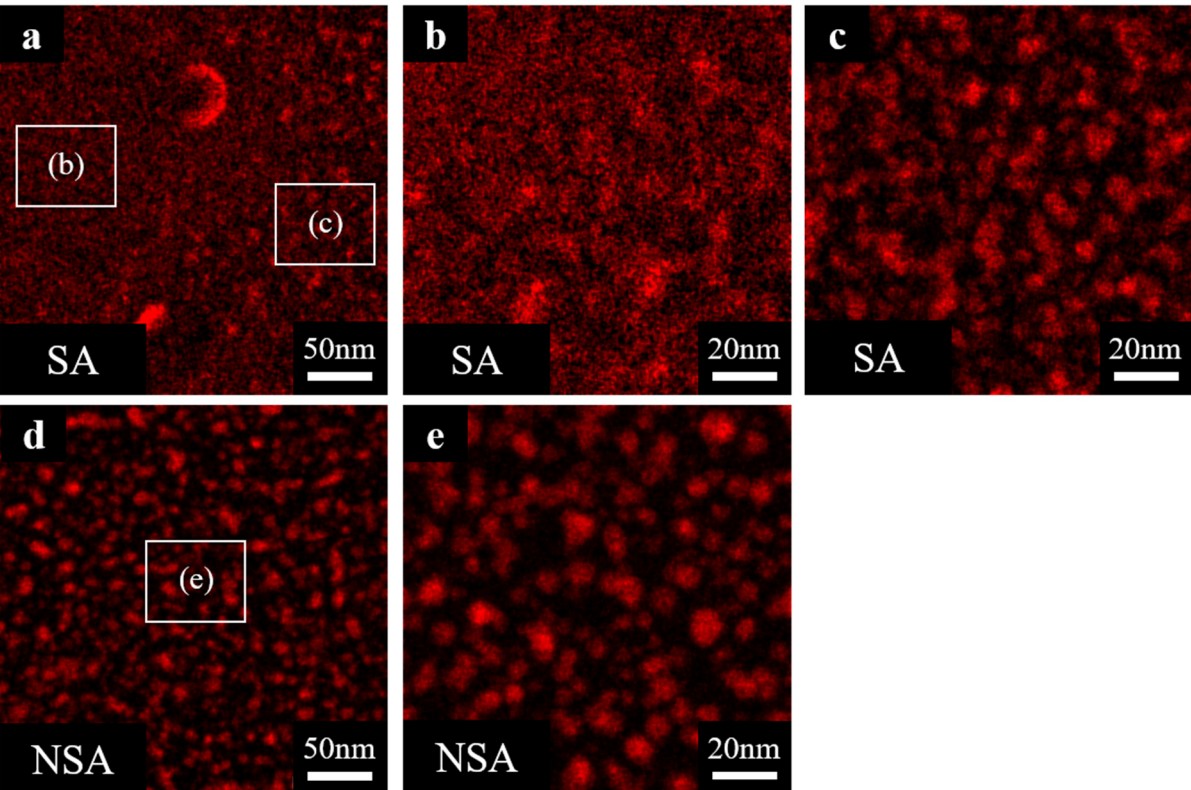

**Figure 8.** Comparison of TEM-EDS analysis of 17-4 PH SS showing nano-scale Cu-rich precipitation.

These results explain the difference in the yield strengths between the SA and the NSA specimens, as shown in Figure 4. The conditions required to form the ideal precipitate Cu particle distribution are considered below.

*3.5. Effects of Normalizing Treatment*

Figure 9 shows the equilibrium phase diagram of 17-4 PH SS simulated using JMatPro software. The results show that the solidification of liquid 17-4 SS begins through the nucleation of primary δ-ferrites at approximately 1430 °C and that the δ-ferrite phase transforms into the austenite phase by solid-state diffusion as the temperature decreases from 1320 to 1120 °C. It is well known that the phase transformation of austenite into martensite by rapid cooling occurs at the starting temperature Ms (132 °C) and ends at the temperature Mf (32 °C) of martensitic transformation [39]. The final structure formed along these paths would then be mostly martensite.

The L-PBF fabrication process of 17-4 PH SS is different and follows the paths of the non-equilibrium solidification processes; thus, it does not agree with the equilibrium state. The simulation results obtained in the equilibrium state predict the solidification and cooling processes and cannot explain the kinetics of the governing phase transformations [24]. At the end of the non-equilibrium solidification process, the non-equilibrium microstructures are still highly likely to exist in the final post-HT state, unless sufficient time and energy are provided at the post-HT stage. Therefore, a kernel average misorientation (KAM) analysis was performed to identify the remaining non-equilibrium phases.

Figure 10 shows the KAM graphs of the SA (Figure 10a) and NSA (Figure 10b) specimens. The KAM graphs quantify the average local misorientation of the nearest neighbors around the measurement point, from which the lattice distortion and dislocation accumulation can be deduced. The results in the KAM graphs are consistent with the larger lattice distortion in the martensite and the smaller distortion in the ferrite microstructure. It was previously reported that the martensite structure of 17-4 PH SS has a high disloca-

tion density, whereas ferrite has a comparatively small dislocation density after solution treatment [17,19].

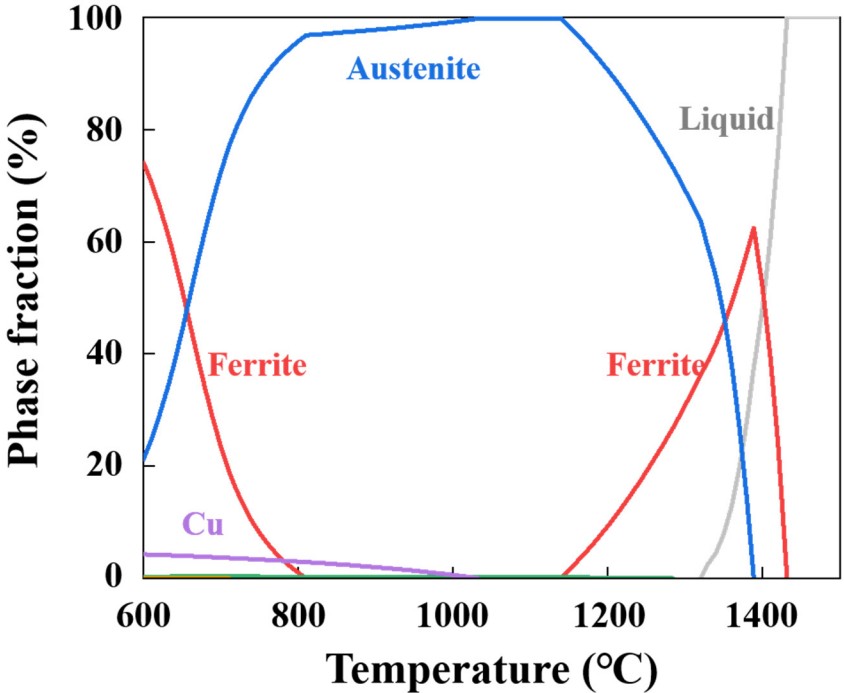

**Figure 9.** Equilibrium phase diagram of 17-4 PH SS calculated by JMatPro software.

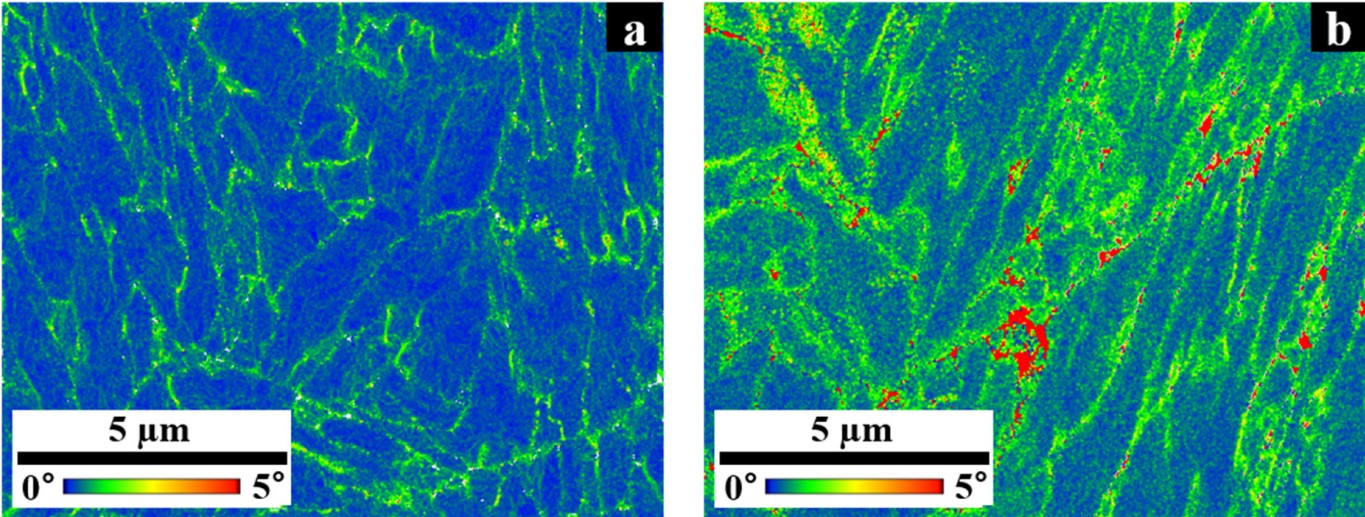

**Figure 10.** Kernel average misorientation graphs of the 17-4 PH steel heat treated by (**a**) SA and (**b**) NSA.

Comparing the SA and NSA specimens, Figure 10a is largely blue (indicating little lattice distortion), and Figure 10b shows larger green and red areas. This means that the SA specimen HT with only the solution process and without the normalizing treatment had a higher probability of including the ferrite phase and not the martensite phase. Conversely, the NSA specimen subjected to both normalizing and solution treatments contained more martensite phases.

The authors suggested that the presence of the ferrite phase after solution HT was caused by insufficient transformation into the solid state owing to the low diffusion rate at 1040 °C [37,40]. Moreover, the 17-4 PH SS powder used in this study had a high Cr

content of 17.10%. Cr acted as a ferrite-stabilizing element, suggesting that more time may be required for solid-state diffusion to transform ferrite into austenite. Nezhadfar et al. [40] confirmed the presence of δ-ferrite after solution HT (1050 °C) and aging HT (482 °C) of L-PBF 17-4 PH SS with high Creq/Nieq values.

Various factors affect Cu precipitation behavior, such as the matrix (δ-ferrite, martensite, and austenite) and microsegregation (ferrite, austenite stabilizer, and precipitation elements). Many researchers have reported that ferrite remains even after HT and that δ-ferrite in 17-4 PH SS has various detrimental effects on hardening behavior [4,17,41].

Sabooni et al. [17] reported that Cu precipitation is delayed in 17-4 PH with a ferrite structure compared to 17-4 PH with a martensite structure. During prolonged aging of the martensite structure, decreases in strength and hardness may occur prematurely through the recovery of dislocations, the coarsening of the Cu-rich precipitates, and the formation of reverted austenite. Additionally, the high dislocation density in the martensitic structure accelerates the precipitation of Cu particles by facilitating pipe diffusion and providing heterogeneous nucleation sites [19].

As shown in Section 3.3, the grain size of 17-4 PH SS fabricated by L-PBF was increased by the normalizing treatment through grain growth and the promotion of Cu precipitation as the ferrite dissolved. It has been reported that the extremely fine-grain size that frequently appears in the AM process lowers the Ms temperature and inhibits the martensitic transformation [42–44]. Colaço et al. [21] reported that the grain size was reduced from 100 to 2.7 μm through the laser surface melting of AISI 420 martensitic SS, and the martensitic transformation temperature decreased from 115 to 20 °C.

Therefore, it can be concluded that the normalizing treatment of L-PBF-fabricated 17-4PH SS facilitates full martensite transformation by dissolving ferrite and triggering grain growth and thereby enabling homogeneous Cu precipitation during the aging HT. Therefore, the mechanical properties of 17-4 PH SS fabricated using L-PBF in a nitrogen gas environment can be improved by adding a normalizing treatment to the conventional process.

## 4. Conclusions

In this study, the effects of the HT profile on the microstructure and mechanical properties of 17-4 PH SS fabricated by L-PBF were studied, and the following conclusions were drawn from the results of a comprehensive series of investigations:

- Significant differences in the mechanical properties of 17-4 PH SS were observed among the various HTs. Compared to the conventional method, the NSA treatment condition resulted in superior mechanical properties that met the ASTM-693 (H900) industrial standard. The yield strength and elongation of the resulting 17-4 PH SS were 1264 MPa and 12.9%, respectively.
- No differences in the austenite fraction were found in the XRD and EBSD results after the additional normalizing treatment. However, irregular fine-grain size and inhomogeneous distribution of the Cu precipitates that affected the mechanical properties of the 17-4 PH SS under SA conditions were observed in the OM and TEM-EDS analyses.
- The existence of a non-equilibrium ferrite phase under the SA condition was observed using KAM, and it was confirmed that the mechanical properties changed based on the differences in the precipitation behavior of the ferrite and martensite structures.
- Cu precipitation in 17-4 PH SS fabricated by L-PBF in a nitrogen environment can be stabilized by applying an additional normalizing treatment. Inhomogeneous Cu precipitation occurs in 17-4 PH SS without a normalizing step because of the mixed ferrite and martensite microstructure.

**Supplementary Materials:** The following are available online at https://www.mdpi.com/article/10.3390/met12050704/s1, Figure S1: EBSD orientation maps and Phase maps (green: ferrite and/or martensite, red: austenite) obtained from the as-built and heat treated 17-4 PH samples: (a-b) AB, (c-d) N, (e-f) S, (g-h) SA, (i-j) NSA; Figure S2: TEM-EDS analysis of SA specimen showing inhomogeneous Cu-rich precipitation (magnification: 450 kx); Figure S3: TEM-EDS analysis of SA specimen showing nano-scale Cu-rich precipitation (magnification: 910 kx); Figure S4: TEM-EDS analysis of SA specimen showing homogeneous Cu-rich precipitation (magnification: 910 kx); Figure S5: TEM-EDS analysis of NSA specimen showing homogeneous nano-scale Cu-rich precipitation (magnification: 450 kx); Figure S6: TEM-EDS analysis of NSA specimen showing homogeneous nano-scale Cu-rich precipitation (magnification: 910 kx); Table S1: Mechanical properties of 17-4 PH SS produced by L-PBF from previous studies [23,26–31]; Table S2: Result of EBSD-Phases map showing phases fraction of AB and HT specimens.

**Author Contributions:** S.-M.Y. and J.Y. wrote and edited the manuscript and contributed to all activities. T.B.K. and S.H.L. fabricated STS 630 parts and conducted experiments and heat treatment. K.C., T.-S.J. and Y.S. contributed to the interpretation and discussion of the experimental results. All authors have read and agreed to the published version of the manuscript.

**Funding:** This research received no external funding.

**Institutional Review Board Statement:** Not applicable.

**Informed Consent Statement:** Not applicable.

**Data Availability Statement:** The data presented in this study are available upon request from the corresponding author.

**Acknowledgments:** This research was performed through R&D projects (Grant No. 20009815 and 20013122), financially supported by the Ministry of Trade, Industry, and Energy (MOTIE) and korea institute of industrial technology internal project (Grant No. EH220013) in Korea. The authors are also grateful to their colleagues for their essential contributions to this work.

**Conflicts of Interest:** The authors declare no conflict of interest.

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
