# Peer review of "Normalizing Effect of Heat Treatment Processing on 17-4 PH Stainless Steel Manufactured by Powder Bed Fusion"

_metals, doi:10.3390/met12050704_

Round 1

Reviewer 1 Report

  1. There are some grammatical errors. The authors should correct all these errors.
  2. In introduction, page 2. It also contains non-equilibrium…. Please unify the representation in phases.
  3. The representation in Fig. 1 is too messy, please revise it.
  4. Please provide image for gas-atomized 17-4 PH SS powders.
  5. The yield strengths of the S and SA specimens were 779 MPa and 722 MPa, respectively,…..? is different from Fig. 1, please confirm it.
  6. In phase analysis, page 8. The former is a fully δ-ferritic BCC microstructure…., and the latter was a mixture of BCC martensitic laths….?
  7. Authors indicated that the phase fractions of all the post-heat-treated specimens were similar. Authors need to quantify the phase fractions of all specimens.
  8. SAED analysis for Cu precipitation is helpful for TEN data in this study. Please provide SAED analysis for Cu precipitation.

Author Response

Dear. Reviewer

Thank you for your review comments.

We have supplemented and corrected our paper for each of the comments.

(Since only one file can be attached, a supplementary and editing certificate are attached at the end of the manuscript.)

We hope our response was an appropriate mention.

We hope you will consider this paper suitable for publication in your journal.

Sincerely,

  1. There are some grammatical errors. The authors should correct all these errors.

It has been proofread our manuscript, and a certificate is attached.

Since only one file can be attached, a supplementary and certificate of editing are attached at the end of the manuscript.

  1. In introduction, page 2. It also contains non-equilibrium. Please unify the representation in phases.

We unified words related non-equillbrium phase as shown below.

- unstanble austenite, retained austenite → retained austenite

- metastable ferrite, δ-ferrite → δ-ferrite

- metastable, non-equillbrium → non-equillbrium

  1. The representation in Fig. 1 is too messy, please revise it.

: We redraw figure 1, and the detailed results are written as a table in the supplementary.

  1. Please provide image for gas-atomized 17-4 PH SS powders.

: We added an image and particle size distribution of gas-atomized 17-4 PH SS powders to figure 2.

  1. The yield strengths of the S and SA specimens were 779 MPa and 722 MPa, respectively,…..? is different from Fig. 1, please confirm it.

: We corrected the value from 779MPa to 593MPa because there was a typing error in the S condition.

  1. In phase analysis, page 8. The former is a fully δ-ferritic BCC microstructure…., and the latter was a mixture of BCC martensitic laths….?

: We corrected the sentence as shown below.

----------------------------------------------------------------------------

“The former is a fully δ-ferritic BCC microstructure formed by the extremely high cooling rate of 105–107 K/s, and the latter was a mixture of BCC martensitic laths and fine FCC austenite grains.”

→ “The columnar grains were a fully δ-ferritic BCC microstructure formed by the extremely high cooling rate of 105–107 K/s, and the fine equiaxed grains were a mixture of BCC martensitic laths and fine FCC austenite grains.

----------------------------------------------------------------------------

  1. Authors indicated that the phase fractions of all the post-heat-treated specimens were similar. Authors need to quantify the phase fractions of all specimens.

: Although it was not considered a significant result in our study, the BCC and FCC phases were written in the supplementary for each condition to help the understanding of the results.

  1. SAED analysis for Cu precipitation is helpful for TEM data in this study.

Please provide SAED analysis for Cu precipitation.

As advised, SAED analysis is a good criterion for deciding Cu precipitation.

TEM specimens are susceptible to contamination, and the specimens we measured are outdated and need to be prepared again.

It was not easy to prepare in a short period for TEM measurement because of the large amount of time and cost in preparing and measuring the specimen.

Our study studied whether the difference in the distribution of chemical components according to the heat treatment conditions affects the mechanical properties through TEM-EDS.

For supplementary explanation, detailed results of Figure 8 were added to the supplement.

There may be differences of opinion, but we hope that the results will supplement your proposal.

Reviewer 2 Report

The manuscript entitled “Normalizing effect of heat treatment processing on 17-4 PH stainless steel manufactured by powder bed fusion” is logically structured and well written. The additive manufacturing for metallic materials is important for industrial applications. Stainless steel is extensively used in various engineering applications. However, the literature review part can be improved to attract the readers from other fields. The Reviewer recommends minor revision after improved literature review. My comments are:

Please introduce additive manufacturing first. Then, give some examples about its wide applications. After that, please highlight the importance of metal additive manufacturing. This will help authors attract other researchers.

Some related works are listed below:

https://dro.deakin.edu.au/view/DU:30152474

http://www.aimspress.com/article/doi/10.3934/mbe.2020255?viewType=HTML

Of course, the authors can add more papers to make a strong literature review.

Author Response

Dear. Reviewer

Thank you for your review comments.

We have supplemented and corrected our paper for the comments.

(Since only one file can be attached, a supplementary and editing certificate are attached at the end of the manuscript.)

We hope our response was an appropriate mention.

We hope you will consider this paper suitable for publication in your journal.

Sincerely,

As shown below, we make up for the introduction with the advantages of the AM process for a complex geometric design part.

----------------------------------------------------------------------------

Recently, higher value-added industries such as aerospace, health care, power generation, and automobiles have demanded more design freedom and complex manufacturing technology to improve functions and performance [5-6].

Additive manufacturing technology has attracted attention as a global solution, with the functional enhancement of dissimilar materials, such as superior corrosion resistance, excellent thermal stability, high hardness, better wear performance, [7-9] and fabrication of near-net-shape parts with higher dimensional accuracy [10].

----------------------------------------------------------------------------

  1. Fu, Y.D. Smooth Topological Design of Continuum Structures for Additive Manufacturing. Deakin University, 2021.
  2. Fu, Y.F. Recent advances and future trends in exploring Pareto-optimal topologies and additive manufacturing oriented topology optimization." Math. Biosci. Eng. 2020, 17.5, 4631-4656.

Reviewer 3 Report

The author made an in-depth study on the effect of heat treatment under different conditions on the properties of the of 17-4 PH SS, and revealed the effect mechanism. This is a good research. However, there are some content that need to be improved. In the introduction section the authors are not very good at pointing out the significance of this study. This study is mainly about the effect of heat treatment conditions on properties of 17-4 PH SS, and has nothing to do with the use of nitrogen or argon in the manufacturing process. In the full manuscript, the author also does not compare the properties of the17-4 PH SS made under nitrogen and the 17-4 PH SS made under argon. Therefore, the authors should not use nitrogen during fabrication as a highlight of this paper. For tensile specimens, heat treated specimens as well as metallographic specimens and EBSD specimens, it is best to attach a picture in the experimental section, indicating where they were taken. I suggest receiving after modification. The authors should perhaps consider following points in the revised submission:

Abstract:

  1. what is the SS? The abbreviation should be interpreted the first time it occurs, not the second time it occurs.

Introduction:

  1. What is As-built (AB) additively manufactured 17-4 PH SS? Rewrite this sentence.
  2. The production of Fig. 1 needs to be improved. It is best to attach a table with data to Fig. 1. Otherwise, it will be difficult for readers to understand. By the way, what is the HT? It means heat treatment? If so, when the HT first appear, need introduce. And what is NA?
  3. “The manufacturing cost of the additive manufacturing process exceeds those of most traditional methods.” Which traditional methods? How much does each traditional method cost, and how much does L-PBF cost? Provide the proof!
  4. I think that some important articles was missed in Introduction. For example: Geng, Y., Konovalov, S.V., Chen, X. Research status and application of the high-entropy and traditional alloys fabricated via the laser cladding (2020) Progress in Physics of Metals, 21 (1), pp. 26-45. DOI: 10.15407/ufm.21.01.026; Xiang, K., Chai, L., Zhang, C., Guan, H., Wang, Y., Ma, Y., Sun, Q., Li, Y. Investigation of microstructure and wear resistance of laser-clad CoCrNiTi and CrFeNiTi medium-entropy alloy coatings on Ti sheet

(2022) Optics and Laser Technology, 145, â„– 107518. DOI: 10.1016/j.optlastec.2021.107518; Natarajan, J., Yang, C.-H., Karuppasamy, S.S. Investigation on microstructure, nanohardness and corrosion response of laser cladded colmonoy-6 particles on 316l steel substrate (2021) Materials, 14 (20), â„– 6183. DOI: 10.3390/ma14206183

Results and Discussion:

  1. “This implies that the austenite phase in the AB and N specimens was transformed to martensite owing to the cooling process during the solution treatment.” The AB specimens without heat treatment. Where is the solution treatment for the AB specimens?
  2. “The authors suggest the large elongated grains were δ-ferrite, and the equiaxed grains retained austenite (Figure 5 (a)).” Add references.
  3. “The SA specimen displays two distinct types of region: 1) Cu particles precipitated homogeneously or 2) no precipitation or only precipitated inhomogeneously. Figure 7 (a) is magnified by 2.5 times in Figures 7 (b) and (c) to reveal the differences more clearly between the two regions.” Rewrite these sentences, change the 1) and 2) to b) and c), it will be good understand.
  4. “The authors suggest the presence of the ferrite phase after solution heat treatment is caused by insufficient transformation into the solid state due to the low diffusion rate at 1040 °C.” Add references.

Author Response

Dear. reviewer

Thank you for your review comments.

We have supplemented and corrected our paper for each of the comments.

(Since only one file can be attached, a supplementary and editing certificate are attached at the end of the manuscript.)

We hope our response was an appropriate mention.

We hope you will consider this paper suitable for publication in your journal.

Sincerely,

The author made an in-depth study on the effect of heat treatment under different conditions on the properties of the of 17-4 PH SS, and revealed the effect mechanism. This is a good research. However, there are some content that need to be improved. In the introduction section the authors are not very good at pointing out the significance of this study. This study is mainly about the effect of heat treatment conditions on properties of 17-4 PH SS, and has nothing to do with the use of nitrogen or argon in the manufacturing process.

In the full manuscript, the author also does not compare the properties of the17-4 PH SS made under nitrogen and the 17-4 PH SS made under argon. Therefore, the authors should not use nitrogen during fabrication as a highlight of this paper. For tensile specimens, heat treated specimens as well as metallographic specimens and EBSD specimens, it is best to attach a picture in the experimental section, indicating where they were taken. I suggest receiving after modification.

: we make up for the introduction section, as shown below.

In previous studies, the effects of mechanical properties of Ar and N atmospheres during the L-PBF process were studied, and it is generally reported that poor mechanical properties appear in a nitrogen atmosphere [12,15,19].

Motivated by this, we studied the microstructure and mechanical properties based on the additional heat treatment and heat treatment steps of 17-4PH SS fabricated under nitrogen conditions, which is advantageous in terms of cost, although has unfavorable mechanical properties. Therefore, heat treatment method for obtaining a stable microstructure and predictable properties is proposed.

The authors should perhaps consider following points in the revised submission:

Abstract:

1. what is the SS? The abbreviation should be interpreted the first time it occurs, not the second time it occurs.

: We corrected the notation order for the first time.

Introduction:

What is As-built (AB) additively manufactured 17-4 PH SS? Rewrite this sentence.

: We corrected the word from As-built (AB) additively manufactured to additively manufactured because of a mistyping.

  1. The production of Fig. 1 needs to be improved.

It is best to attach a table with data to Fig. 1.

Otherwise, it will be difficult for readers to understand.

By the way, what is the HT? It means heat treatment? If so, when the HT first appear, need introduce.

And what is NA?

: We redraw figure 1, and the detailed results are written as a table in the supplementary.

  1. “The manufacturing cost of the additive manufacturing process exceeds those of most traditional methods.” Which traditional methods? How much does each traditional method cost, and how much does L-PBF cost? Provide the proof!

: In many studies, the disadvantages of the L-PBF process in terms of cost and manufacturing time have been known, and a reference for this has been added.

  1. Wilson, J.M.; Piya, C.; Shin, Y.C.; Zhao, F.; Ramani, K. Remanufacturing of turbine blades by laser direct deposition with its energy and environmental impact analysis. J. Clean. Prod. 2014, 80, 170–178.
  2. Miedzinski, M. Materials for Additive Manufacturing by Direct Energy Deposition. Master ’s Thesis, Chalmers University of Technology, Gothenburg, Sweden, 2017.
  3. DebRoy, T.; Wei, H.L.; Zuback, J.S.; Mukherjee, T.; Elmer, J.W.; Milewski, J.O.; Beese, A.M.; Wilson-Heid, A.; De, A.; Zhang, W. Additive manufacturing of metallic components-Process structure and properties. Prog. Mater. Sci. 2018, 92, 112–224.
  4. Doubenskaia, M.; Domashenkov, A.; Smurov, I.; Petrovskiy, P. Study of selective laser melting of intermetallic TiAl powder using integral analysis. Int. J. Mach. Tools Manuf. 2018, 129, 1–14.
  5. Shamsaei, N.; Yadollahi, A.; Bian, L.; Thompson, S.M. An overview of direct laser deposition for additive manufacturing: Part 2: Mechanical behavior, process parameter optimization and control. Addit. Manuf. 2015, 8, 12–35.

  1. I think that some important articles was missed in Introduction.

We make up for the introduction with the advantages of the AM process for parts of functional enhancement, as shown below.

Recently, higher value-added industries such as aerospace, health care, power generation, and automobiles have demanded more design freedom and complex manufacturing technology to improve functions and performance [5-6].

Additive manufacturing technology has attracted attention as a global solution, with the functional enhancement of dissimilar materials, such as superior corrosion resistance, excellent thermal stability, high hardness, better wear performance, [7-9] and fabrication of near-net-shape parts with higher dimensional accuracy [10].

  1. Geng, Y.; Konovalov, S.V.; Chen, X. Research status and application of the high-entropy and traditional alloys fabricated via the laser cladding. Usp. Fiz. Met. 2020, 21(1), 26-45.
  2. Xiang, K.; Chai, L.; Zhang, C.; Guan, H.; Wang, Y.; Ma, Y.; Sun, Q.; Li, Y. Investigation of microstructure and wear resistance of laser-clad CoCrNiTi and CrFeNiTi medium-entropy alloy coatings on Ti sheet. Opt. Laser. Technol. 2022, 145, 107518.
  3. Natarajan, J.; Yang, C.-H.; Karuppasamy, S.S. Investigation on microstructure, nanohardness and corrosion response of laser cladded colmonoy-6 particles on 316l steel substrate. Materials, 2021, 14(20), 6183.

  1. Results and Discussion:

“This implies that the austenite phase in the AB and N specimens was transformed to martensite owing to the cooling process during the solution treatment.” The AB specimens without heat treatment. Where is the solution treatment for the AB specimens?

: We thought that the expression caused a misunderstanding, so the sentence was corrected.

“This implies that the austenite phase in the AB and N specimens was transformed to martensite owing to the cooling process during the solution treatment.”

→ “This implies that, during cooling after solution heat treatment, the FCC phase in AB and e specimens was transformed into BCC phase.”

  1. “The authors suggest the large elongated grains were δ-ferrite, and the equiaxed grains retained austenite (Figure 5 (a)).” Add references.

: We added a reference to the morphology of the 17-4 PH SS phase as shown follows.

  1. Zai, L.; Zhang, C.; Wang, Y.; Guo, W.; Wellmann, D.; Tong, X.; Tian, Y. Laser powder bed fusion of precipitation-hardened martensitic stainless steels: a review. Metals, 2020, 10(2), 255.

  1. “The SA specimen displays two distinct types of region: 1) Cu particles precipitated homogeneously or 2) no precipitation or only precipitated inhomogeneously. Figure 7 (a) is magnified by 2.5 times in Figures 7 (b) and (c) to reveal the differences more clearly between the two regions.” Rewrite these sentences, change the 1) and 2) to b) and c), it will be good understand.

: We have revised the notation of your suggestion from the 1) and 2) to (b) and (c)

  1. “The authors suggest the presence of the ferrite phase after solution heat treatment is caused by insufficient transformation into the solid state due to the low diffusion rate at 1040 °C.” Add references.

: We added reference to the effect of solution heat treatment on the phase as shown below. 

  1. Zhao, Z.; Wang, H.; Huo, P.; Bai, P.; Du, W.; Li, X.; Li, J.; Zhang, W. Effect of Solution Temperature on the Microstructure and Properties of 17-4PH High-Strength Steel Samples Formed by Selective Laser Melting. Metals. 2022, 12, 425.
  2. Nezhadfar, P. D.; Burford, E.; Anderson-Wedge, K.; Zhang, B.; Shao, S.; Daniewicz, S. R.; Shamsaei, N. Fatigue crack growth behavior of additively manufactured 17-4 PH stainless steel: Effects of build orientation and microstructure. Int. J. Fatig. 2019, 123, 168-179.

Round 2

Reviewer 3 Report

Authors have made correcton. No new comments.